# The Function and Regulation Mechanism of Non-Coding RNAs in Muscle Development

**DOI:** 10.3390/ijms241914534

**Published:** 2023-09-26

**Authors:** Yaling Yang, Jian Wu, Wujun Liu, Yumin Zhao, Hong Chen

**Affiliations:** 1College of Animal Science, Xinjiang Agricultural University, Urumqi 830052, China; yangyaling3141@126.com (Y.Y.); lwj-ws@sohu.com (W.L.); 2Key Laboratory of Beef Cattle Genetics and Breeding, Ministry of Agriculture and Rural Affairs, Academy of Agricultural Sciences of Jilin Province, Changchun 136100, China; wujian0303@126.com

**Keywords:** muscle development, miRNA, lncRNA, circRNA

## Abstract

Animal skeletal muscle growth is regulated by a complex molecular network including some non-coding RNAs (ncRNAs). In this paper, we review the non-coding RNAs related to the growth and development of common animal skeletal muscles, aiming to provide a reference for the in-depth study of the role of ncRNAs in the development of animal skeletal muscles, and to provide new ideas for the improvement of animal production performance.

## 1. Introduction

Skeletal muscle is an important part of the animal body, which accounts for about 40% of the animal’s body weight, and plays an important role in the body’s metabolism, force generation and locomotion [1,2]. In addition, in livestock production, the growth and development of skeletal muscle determines the animal’s ability to produce meat, and slow or abnormal myogenesis can adversely affect the livestock economy; thus, it is crucial to study the growth and development of animal skeletal muscle. In recent years, with the advancement of high-throughput sequencing technology and the development of bioinformatics, a large number of noncoding RNAs (ncRNAs) have been identified in domestic animals to regulate skeletal muscle cell development, mainly including microRNAs (miRNAs), long noncoding RNAs (lncRNAs), and circular RNAs (circRNAs) [3]. Non-coding RNAs contribute significantly to organismal complexity by diversifying eukaryotic gene regulatory mechanisms [4]; many in vitro studies, including overexpression and knockdown, have been applied to study the role of non-coding RNAs [5]; In addition, mouse models help to better understand their role in muscle development [6]. Identification of a large number of non-coding RNAs as key regulators of muscle growth and development in animals has been gradually increasing our understanding of their role. This review focuses on the regulation of animal skeletal muscle development by miRNAs, lncRNAs and circRNAs, aiming to provide a deeper understanding of the mechanisms by which non-coding RNAs regulate the growth and development of animal skeletal muscle.

## 2. Overview of Skeletal Muscle Growth and Development in Animals

The basic structure of skeletal muscle consists of three main types of cells: myocytes, adipocytes, and fibroblasts [7]. Skeletal muscle originates from progenitor cells in vertebrate limb somites [8]. Myogenesis is divided into two stages. The first stage refers to the formation of myoblasts through complex changes in progenitor cells during embryonic development, which proliferate, migrate, and fuse to form multinucleated myotubes induced by specific myogenic transcription factors; the myotubes ultimately fuse to become myofibers. Myogenic cells are the precursors of myoblasts, and their proliferation and differentiation determine the growth and development of skeletal muscle [9,10]. This stage of muscle development determines the number and structure of muscle fibers in domestic animals [11]. In phase II, postnatal muscle growth and development is closely related to the proliferation and differentiation of stem and progenitor cells (satellite cells) [12,13,14]. In addition, animal skeletal muscle is a heterogeneous tissue whose function depends on myotube growth and differentiation, transformation of myofiber types, and mitochondrial functions [15,16,17]. This implies that the process of myogenesis requires the coordination of multiple factors to control the activation of quiescent satellite cells, the proliferation of adult myoblasts, exit from the cell cycle, and subsequent terminal differentiation leading to multinucleated myofibers [18,19].

Skeletal muscle maintains respiratory, locomotor and metabolic functions in animal life activities, and a large number of transcription factors are involved in the regulation of skeletal muscle formation, such as myogenic regulatory factors (MRFs) and myocyte enhancer factor 2 (MEF2) [20,21]. MRFs whose families include myogenic differentiation (MyoD), myogenin (MyoG), myogenic factor 5 (Myf5) and myogenic factor 6 (Myf6), especially MyoD and MyoG are essential for myogenic progenitor cell assays and muscle cell proliferation and differentiation, and MyoG is required for assembly of the transcription machinery on muscle genes during skeletal muscle differentiation [22,23]. In addition, transcription factors PAX3 and PAX7 from the *PAX* gene family are the major groups in the early stages of myogenesis [24,25]; MEF2 belongs to the MADS supergene family, which includes MEF2A, MEF2B, MEF2C and MEF2D, and can act synergistically with members of the MRF family mentioned above to promote or inhibit skeletal muscle growth and differentiation in animals [26,27,28]. These transcription factors combine to form a complex and diverse system of signaling pathways that regulate the mechanisms of muscle production and influence the growth and development of muscle cells.

## 3. ncRNAs and Their Mechanisms of Action

Non-coding RNAs are a class of RNAs transcribed from genomic loci that have no translational function but are capable of receiving or transmitting regulatory signals. These enigmatic molecules play important roles in biology and have a variety of biological functions such as controlling chromosome dynamics, splicing, RNA editing, translational repression, and the molecular mechanisms of mRNA destruction [29]. They can regulate transcription through chromatin interactions, chelate miRNAs and proteins and the post-translational modification of proteins, and mediate mRNA translation and stability [30,31]. Currently, the most commonly studied noncoding RNAs are microRNAs (miRNAs), long noncoding RNAs (lncRNAs), and circular RNAs (circRNAs). An overview of the biology and dynamics of noncoding RNAs is shown in Figure 1.

MicroRNAs are a class of small non-coding RNAs, consisting of approximately 19–25 nucleotides, which often regulate gene expression post-transcriptionally and play an important role in processes such as cell proliferation and metabolic stabilization [32]. MiRNAs are initially transcribed into primitive miRNA transcripts (pri-miRNAs), which are then cleaved into approximately 70 nt of precursor miRNAs (pre-miRNAs) by an RNase III enzyme.) The precursor miRNAs are exported from the nucleus to the cytoplasm and cleaved again by the Dicer enzyme to form miRNA double-stranded bodies containing mature miRNA strands [33]. Mature miRNAs are mainly able to bind to the complementary sites in the 3′-UTR region of the mRNA of target genes, triggering degradation or translational repression of the target mRNAs, and some of the miRNAs can also bind to the 5′-UTR and coding region, thus exerting silencing effects on gene expression [34,35]. Conversely, a few other studies have found that miRNAs can promote gene expression [36,37].

Long noncoding RNA is a transcript that does not code for protein and is greater than 200 nucleotides in length [38]. Its sequence conservation and expression levels are lower than those of protein-coding genes, but a growing body of evidence confirms that it plays a key role in biological processes such as organ growth and development [39], cell proliferation and differentiation [40], glycolipid metabolism [41], and disease [42]. This is mainly due to the intrinsic nature of RNA molecules and their ability to form complex secondary structures that allow them to bind a wide range of molecules. Most lncRNAs are transcribed from RNA polymerase II and have mRNA-like features such as 5′ caps, polyA tails and splice sites. In addition, lncRNAs can be categorized into the following four groups based on their genomic location and environment: (1) intergenic lncRNAs (lincRNAs); (2) intronic lncrna; (3) lncrna with overlapping significance, and (4) antisense lncrna [43,44]. Lncrna can be functionally subdivided into the following three subclasses based on its mode of action: (1) functioning at the transcriptional level; (2) acting as a decoy for certain molecules, such as transcription/splicing factors in the nucleus, miRNAs in the cytoplasm, or RNA degradation complexes, to regulate gene expression levels [45]; and (3) serve as scaffolds for the formation of complex molecular mechanisms or nuclear subdomains that influence the level of gene expression [46].

Circular RNA is a class of covalently closed circular RNA molecules that do not have a 5′ end cap and 3′ end poly tail structure [47], They are abundantly and widely expressed in eukaryotes, with interspecies conservation, stable expression and spatiotemporal specificity. circRNAs can be generated through different circularization mechanisms and sources to produce six types of circRNAs, namely exonic circRNAs (ecircRNAs), intronic circular RNAs (ciRNAs), and exon-intronic circRNAs (EIciRNAs), tricRNA formed during tRNA processing, f-circRNA of fusion gene origin and MecciRNA of mitochondrial origin [48,49,50]. The role of circRNAs in influencing gene expression at multiple levels and their mechanisms mainly include transcriptional regulation, adsorption of miRNAs, binding/translation of proteins, and insertion into the genome as pseudogenes after reverse transcription [51,52]. In addition, a large number of studies have also shown that these three RNAs are widely involved in various important life activities in animals, including organ growth and development, cell proliferation and differentiation, glycolipid metabolism, and disease aspects, etc. [39,53,54,55].

## 4. Regulation of Skeletal Muscle Growth and Development by ncRNAs

### 4.1. Regulation of Skeletal Muscle Growth and Development by miRNAs

miRNAs have spatiotemporal expression patterns, which form a very fine network of regulatory mechanisms in organisms, and they cause dynamic changes in different tissues at different times. A single mRNA can be regulated by multiple miRNAs, and a single miRNA can likewise target multiple mRNAs [56]. In recent years, miRNAs have played important roles in animal skeletal muscle development, including regulation of myoblast proliferation, differentiation, fusion, apoptosis and fiber-type specification [57]. Among these, most miRNAs mainly affect the proliferation and differentiation of myoblasts and satellite cells (see Table 1 and Figure 2).

Both adult muscle regeneration and embryonic myogenesis share a similar genetic hierarchy, arranged by a cascade of myogenesis transcription factors, including the previously mentioned transcription factors Pax3/7, MRFs and MEF2) [20,21,24,25]. These transcription factors can act synergistically or antagonistically with each other. miRNAs can target multiple myogenesis transcription factors and thus affect myogenesis; at the same time, their expression is directly controlled by these transcription factors [83,84]. This relationship between miRNAs and myogenesis regulatory factors supports the fact that miRNAs are integral to myogenesis regulatory network components. Several miRNAs have been identified as direct repressors of myogenesis transcription factors that are capable of controlling the process of myogenesis differentiation. For example, miR-2425-5p targets the 3′-UTR of *myogenin*, thereby inhibiting satellite cell differentiation [67]. Similarly, miR-885 inhibits cell differentiation by suppressing MyoD activity [68]. 

#### 4.1.1. miRNAs Are Involved in Skeletal Muscle Development in Satellite Cells

Satellite cells (SMSCs) are muscle stem cells with the ability to proliferate and differentiate into muscle fibers when the muscle is damaged but which remain quiescent under normal conditions due to the expression of paired frame transcription factor (Pax7). Once subjected to stress, such as heavy loading, trauma, etc., the expression of Pax7 and MyoD changes, causing the cells to enter a proliferative phase and produce a large number of myofibroblasts. Myofibroblasts further differentiate and fuse to form myotubes by increasing the expression of MRFs, which promotes skeletal muscle repair and regeneration [85,86]. miRNA helps to keep SMSCs dormant and also regulates skeletal muscle proliferation, differentiation and apoptosis, which are essential for skeletal muscle regeneration. The miRNAs identified so far that act on SMSC proliferation and differentiation in domestic chickens include miR-99a-5p, miR-21-5p and miR-148a-3p. The mechanism of action of miR-99a-5p is as follows: miR-99a-5p targets the myotubularin-related protein 3 (*MTMR3*), stimulates the proliferation of SMSCs, and inhibits the differentiation of SMSCs [58]. miR-21-5p promotes the proliferation and differentiation of SMSCs by targeting Krüppel-like factor 3 (*KLF3*) [59]. miR-148a-3p down-regulates mesenchymal homology box 2 (*Meox2*), activates the PI3K/AKT signaling pathway, promotes SMSC differentiation, and inhibits apoptosis [60]. In calves this includes miR-377, miR-23a, and miR-2425-5p, and the mechanism of action is as follows: miR-377 targets *FHL2* and attenuates the Wnt/β-catenin signaling pathway to inhibit the proliferation and differentiation of SMSCs [65]. miR-23a promotes myogenic differentiation of fetal bovine skeletal muscle-derived progenitor cells by targeting the MyoD family inhibitor structural domain-containing (*MDFIC*) genes [66]. miR-2425-5p targets the 3′-UTR of RAD9 homolog A (*RAD9A*) and myogenesis (*MYOG*) to promote proliferation and inhibit differentiation [67] in sheep and goats, including miR-181a, miR-29a, miR-99b-3p, and miR-27a-3p [69,70,71,72] and in rabbits and pigs, including miR-194-5p and miR-22, respectively [80,81]. The specific mechanisms of action of these miRNAs will not be elaborated. See Table 1.

#### 4.1.2. MiRNA Modulates Skeletal Muscle through Regulating Myoblasts

Myoblasts are one of the most important components of embryonic muscle development and endogenous repair [87]. Many studies have shown that miRNAs affect the growth and differentiation of animal myogenic cells in vitro. Myogenesis-associated miRNAs (miR-1a-3p, miR-133a-3p, miR-133b-3p, miR-206-3p, miR-128-3p, miR-351-5p) negatively regulate the NK/MAPK signaling pathway, which suppresses skeletal muscle differentiation [88]. Among these, miR-1, miR-133, and miR-206 are three highly conserved muscle-specific miRNAs [89], and local injection of muscle-specific miRNA accelerates muscle regeneration in a rat model of skeletal muscle injury [90]. miR-424-5p targets the heat shock protein 90α family class A member 1 (*HSP90AA1*) and also promotes proliferation and inhibits differentiation during mouse skeletal muscle development [75]. In poultry, miR-7 targets the Krüppel-like factor 4 (*KLF4*) and inhibits the proliferation and differentiation of myoblasts [61]. miR-214 targets the tRNA methyltransferase 61A (*TRMT61A*) gene to inhibit myofibroblast proliferation and promote their differentiation [63]. It was also found that miR-2954 blocked yin-yang 1 (*YY1*), promoted myofibroblast proliferation, and inhibited differentiation [62]. miR-29b-1-5p targets the anchoring protein repeat domain 9 (*ANKRD9*), inhibits myoblast proliferation, and promotes their differentiation [64]. miRNAs were found to affect the growth of myofibroblasts in cattle, sheep, and ducks, including miR-885, targeting *MyoD1* to promote proliferation and inhibit differentiation of bovine myofibroblasts [68]. miR-22-3p targets *IGFBP3* in hu sheep to inhibit proliferation and promote differentiation of skeletal myoblasts [74]. miR-145-5p targets the coding structural domain sequence of *USP13* in goats to inhibit myoblast differentiation [73]. miR-1 targets histone deacetylase 4 (*HDAC4*) in ducks to promote myoblast differentiation, whereas miR-133 affects serum response factor (*SRF*) and transforming growth factor β receptor 1 (*TGFBR1*) expression to promote myoblast proliferation [82]. The fusion of myoblasts with myotubes is a critical stage in the formation of skeletal muscle, and the fusion process in myogenesis has been extensively studied in recent years. In domestic chickens, miR-140-3p partially inhibits *Myomaker* expression in vitro by binding to the 3′ UTR of *Myomaker* to inhibit myofibroblast fusion [91]. In mice, miR-96-5p targets the 3′ UTR of the four-and-a-half LIM domain 1 (*FHL1*) mRNA to inhibit myoblast differentiation and fusion [92]. Apoptosis of myoblasts is an important process in myogenesis, and inhibition of apoptosis in adult myoblasts can lead to impaired skeletal muscle development, abnormalities, inflammation, and tumorigenesis. In poultry, miR-146b-3p was found to inhibit *MyoD* family inhibitor-containing structural domain (*MDFIC*) and PI3K/AKT pathways to promote apoptosis in chicken embryo myoblasts [93]. In cattle, miR-652 was found to target the insulin enhancer binding protein-1 (*ISL1*) to promote proliferation and differentiation and inhibit apoptosis in yak myoblasts [94].

In conclusion, miRNAs are involved in almost every step of myogenesis, and as the miRNA field enters a more mature stage, more and more muscle-specific miRNA regulatory mechanisms are bound to be discovered.

#### 4.1.3. Specification and Maintenance of Fiber Type-Associated miRNAs

All mammals contain a variety of different skeletal muscle subtypes, and different muscle-fiber types affect quality characteristics such as meat color, tenderness, water retention, juiciness, and flavor [95]. Typically, muscle fibers are divided into type I (slow) and type II (fast) fibers, with type II fibers consisting of three subtypes, IIa, IIb, and IIX [96]. Much remains unclear about the molecular regulation of different muscle-fiber types in animals, with fewer examples involving ncRNAs. In domestic chickens, Liu et al. found that MiR-499-5p targets a repressor of slow muscle-specific gene expression (*SOX6*), whereas MiR-196-5p targets cmp-pkg and a key component of the calcium signaling pathway (*CALM1*), which together regulate slow muscle fiber formation [97]. In mouse skeletal muscle, overexpression of miR-499 can completely transform fast muscle fibers of soleus muscle into slow muscle fibers [98]. In contrast, double knockout of miR-499 and miR-208b resulted in a significant loss of type I fibers in the soleus [98].

### 4.2. Regulation of Skeletal Muscle Growth and Development by lncRNA

LncRNAs can bind to miRNAs and affect skeletal muscle proliferation and differentiation. lncRNA-MEG3 can promote bovine skeletal muscle differentiation by interacting with miRNA-135 and myocyte enhancer factor 2C (MEF2C), suggesting that it plays an important role in the process of myogenesis [99]. In addition, some lncRNAs regulate gene expression in cis or trans; e.g., MUNC lncRNA acts as an enhancer RNA for the pro-muscle factor *Myod1* gene in a cis-regulatory manner, which can stimulate the expression of other pro-muscle genes in trans by recruiting the cohesin complex, thereby regulating skeletal muscle growth [100]. In addition, some lncRNAs can affect muscle development and atrophy by modifying proteins. lncFAM interacts with the myogenic protein MYBPC2 and recruits the RNA-binding protein HNRNPL as the promoter of MYBPC2, which increases the transcription of the MYBPC2 mRNA and promotes the production of MYBPC2, promoting myogenesis [101]. Although lncRNA has been associated with skeletal muscle development in pigs [102], cattle [103], sheep [104], goats [105], and domestic chickens [106], fewer studies have been conducted on its mechanisms. According to existing studies, the roles of LncRNAs mainly in animal skeletal muscle include acting as cis- or trans-regulators or regulating gene expression through sponge competition for miRNA-encoded short molecular micropeptides. Therefore, we summarize the expression and functions of LncRNAs in the skeletal muscles of some animals as shown in Table 2.

#### 4.2.1. lncRNAs Regulate Skeletal Muscle through Sponge miRNAs

Emerging studies have shown that lncRNAs can regulate muscle proliferation and differentiation by competing for endogenous RNAs. lncRNA-MEG3 can act as a molecular sponge for miR-133a-3 p to regulate the expression level of *PRRT 2*, thereby modulating skeletal muscle regeneration [107]. In pigs, lncRNA H19 can act as a molecular sponge for miR-140-5p to inhibit porcine skeletal muscle satellite cell differentiation and can also bind directly to Drebrin 1 to regulate satellite cell differentiation [108]. In mice, lncRNA H19 can regulate *TGFBR2* expression through endogenous competing RNA functions, enabling miR-20a-5p to activate the TGFβ/Smad pathway and promote skeletal muscle fibrosis in mice [109]. Furthermore, lncMFAT1 can act as a miR-135a-5p sponge to activate the TGFBR135/SMAD5 pathway and promote skeletal muscle fibrosis in mice [110]. In cattle, lncA2B1 can regulate *HNRNPA2B1* expression through endogenous competing RNA functions, enabling miR-206 to promote bovine myoblast differentiation and myogenesis [111]. miR-133a-targeted *GosB* is sponged by muscle differentiation-associated lncRNA (*MDNCR*) to promote the differentiation of bovine primary myoblasts [115]. In goats and sheep, lncR-133a acts as a miR-133a-3p sponge, activates the FGFR1/ERK1/2 pathway in goats, and inhibits myogenic cell differentiation [112]. lncRNA (*CTTN-IT1*) can target miR-29a to act as a competing endogenous RNA for *YAP1* and promote the proliferation and differentiation of skeletal muscle satellite cells in hu sheep [106].

#### 4.2.2. LncRNAs Regulate Skeletal Muscle through Cis or Trans Gene Expression

In mice, lncMyolinc controls the expression of the protein-coding gene *Filip1* in a cis manner, knockdown of lncMyolinc and *Filip1* inhibits the differentiation of myoblasts into myotubes, and *Myolinc* binds to the TAR DNA-binding protein (TDP-43) during myogenesis, resulting in the expression of muscle-specific genes [113]. In the chicken, lncRNA-Six1 overexpression enhances the expression of muscle growth-related genes (*MYOG*, *MYHC*, *MYOD*, *IGF1R*, and *INSR*) and encodes a micropeptide that affects the expression of the Six1 protein in a cis-regulatory manner and promotes the proliferation of myogenic cells [114]. In bovine, lnc403 negatively regulates the expression of the neighboring gene *Myf6* and can positively regulate the expression of the interacting protein *KRAS*, inhibiting bovine skeletal muscle satellite cell differentiation [116].

### 4.3. Regulation of Skeletal Muscle Growth and Development by circRNA

#### 4.3.1. CircRNA Regulates Skeletal Muscle Growth through Sponge miRNAs

CircRNAs are associated with biological processes such as cell proliferation, survival, and differentiation, and are widely and stably expressed in eukaryotes with tissue and cell specificity [117]. CircRNAs are abundant in skeletal muscle and are involved in myogenesis. Table 3 lists some of the recently discovered mechanisms by which circRNAs regulate the growth and development of skeletal muscle in animals. circMYL1 can act as a molecular sponge for miR-2400 in cattle, inhibiting the proliferation of bovine adult myoblasts and promoting differentiation [118]; adsorption of miR-411a by circNDST1 attenuates the inhibitory effect on its target gene *Smad4*, thereby promoting proliferation and inhibiting cell differentiation in bovine myoblasts [119]. In domestic chickens, circITSN2 can target miR-218-5p, thereby attenuating the inhibitory effect on *LMO7* and promoting skeletal muscle development in chicken embryos [120]; circCCDC91 can bind to the miR-15 family, promote the proliferation and differentiation of myoblasts and attenuate skeletal muscle atrophy by activating the activated IGF1-PI3K/AKT pathway [121]. In goats, circUSP13 adsorbs miR-29c to target and regulate *IGF1* to promote differentiation and inhibit apoptosis in goat myoblasts [122]; circUBE3A acts as a miR-28-5p sponge to promote myogenic cell proliferation and differentiation [123]. In pigs, circIGF1R targets miR-16 and promotes myoblast differentiation [124]. These processes suggest that circRNAs can play an important function in regulating muscle development by acting as molecular sponges for miRNAs to inhibit miRNA activity.

#### 4.3.2. CircRNA Is Involved in Different Growth and Developmental Processes in Skeletal Muscle

Different circRNAs can be involved in different growth and developmental processes in skeletal muscle. A total of 481 differentially expressed circRNAs were identified from skeletal muscle samples from embryos, rabbits, and adult rabbits, of which the highest expressed, novel_circ_0025664, was found to be expressed only in adulthood [126]. circEDC3 was differentially expressed at all four time points of thoracic muscle development in broiler- and laying-hen embryos, and validation of circEDC3 in thoracic and leg muscles of different embryonic ages revealed that the expression level of circEDC3 decreased significantly with embryonic age [127]. The circRNAs were also dynamically expressed in different embryonic muscles of pigs, and some of them were significantly down-regulated with the growth and development of embryonic skeletal muscles [128]. This dynamic expression pattern reveals the unique regulatory roles played by different circRNAs in various stages of animal skeletal muscle development: it may affect gene expression and protein synthesis in a variety of ways, thus playing an important role in skeletal muscle cell proliferation, differentiation, and function.

#### 4.3.3. CircRNAs Are Directly Converted into Proteins Involved in Skeletal Muscle Development

Circular RNAs may be translated into functional polypeptides in the presence of internal ribosome entry sites (IRES) and open reading frames (ORFs). Based on the number of open reading frames and N6-methyladenosine motifs, the screen identified 224 circRNAs with coding potential, while GO and KEGG analyses of the target genes of 75 known circRNAs predicted that differentially expressed circRNAs might be involved in skeletal muscle development [129]. In domestic chickens, Circ-FAM188B is a circRNA molecule with protein-coding capacity, and bioinformatic tools predicted that Circ-FAM188B contains an ORF with the potential to encode Circ-FAM188B-103aa, which was confirmed by the presence of IRES. Circ-FAM188B promotes the proliferation of chicken satellite cells but inhibits their differentiation [130]. In addition, circRNAs can form RNA-protein complexes to fulfill their functions, e.g., circFOXK2 can directly interact with 94 proteins [131]; circSamd4 binds to PUR proteins and inhibits the antagonistic effect of PUR proteins on the transcription of MHC genes, which promotes the differentiation of adult myoblasts and accelerates the process of skeletal muscle growth and development [132]. CircMYBPC1 directly binds MyHC proteins and promotes myoblast differentiation and skeletal muscle regeneration [133] and circSVIL can interact with STAT1 to promote proliferation and inhibit apoptosis in bovine myoblasts [134].

## 5. Conclusions

Skeletal muscle is an important tissue in animals, and skeletal muscle growth and biology are closely related to meat production, so a full understanding of the role of non-coding RNAs in the development of animal skeletal muscle can better control the growth rate and quality of livestock and poultry, as well as improve their economic efficiency. At the same time, it is of great significance to promote the development of modern agriculture. Currently, researchers have identified a large number of ncRNAs that regulate skeletal muscle in animals such as pigs, bovine, sheep, chickens, and mice, among which the mechanism of miRNA regulation of skeletal muscle development has been more clearly understood. lncRNAs are more difficult to study because of their low sequence conservation among species, and the advanced structure of lncRNA itself affects their mode of action; most circRNAs are interspecies conserved but their research methods and role in skeletal muscle development still need further exploration and improvement. Currently, miRNAs as circRNA and lncRNA sponges are one of the most studied modes of regulation. Much remains unknown about the specific mechanisms by which non-coding RNAs act in skeletal muscle development, and more in-depth excavation of the network of interactions between ncRNAs and other molecules and how they synergistically regulate the entire cellular system is needed to advance the field.

## Figures and Tables

**Figure 1 ijms-24-14534-f001:**
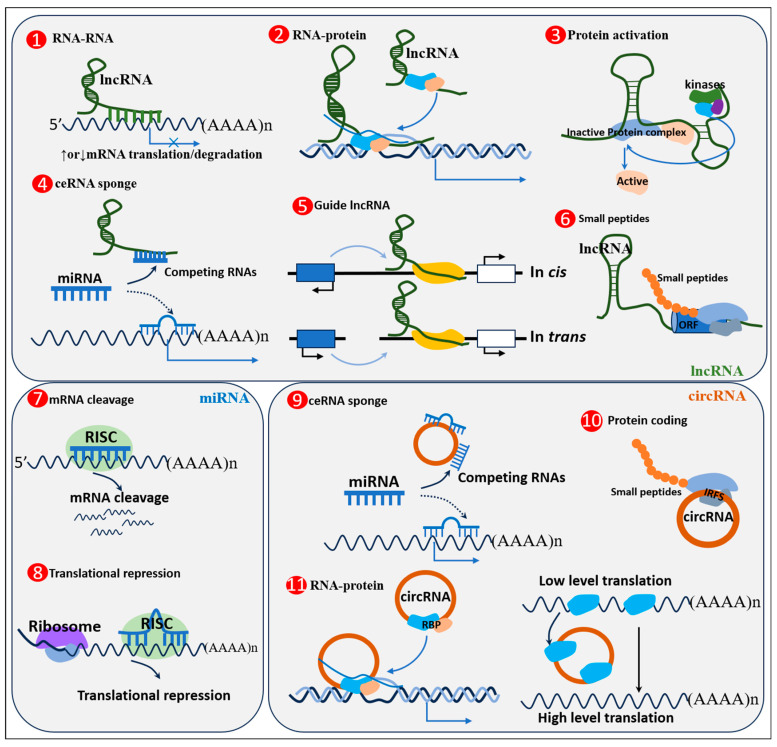
Biological and mechanical overview of non-coding RNAs. (**1**) LncRNA can combine with mRNA pairing to influence translation, variable shearing and the stability of the mRNA. (**2**) Recruitment of transcription factors to promote transcription of target genes or recruitment of chromatin modifiers. (**3**) Acting as scaffold protein interactions or bridges, affecting the formation of protein polymers, regulating the activity of protein. (**4**) As molecular sponges, lncRNAs competitively adsorb miRNAs and reduce their impact on mRNA transcription and translation. (**5**) lncRNAs can regulate gene expression in a cis or trans manner. (**6**) Part of the lncRNAs have the function of the translation of small peptide. (**7**) The miRISC complex (Ago2-microRNA) target mRNA by either perfect complementarity, producing transcript degradation, or (**8**) an imperfect one promoting translation repression. (**9**) CircRNA as endogenous competitive RNA regulation of miRNAs affect transcription process. (**10**) CircRNA translates small peptides to regulate physiological processes. (**11**) The functions of circRNAs are favoring or inhibiting polysome loading to mRNAs, acting as a decoy to preclude access of regulatory proteins to DNA (**left side**) or mRNA (**right side**). Solid arrows indicate that transcription or translation processes are promoted, and dashed arrows indicate competitive repression.

**Figure 2 ijms-24-14534-f002:**
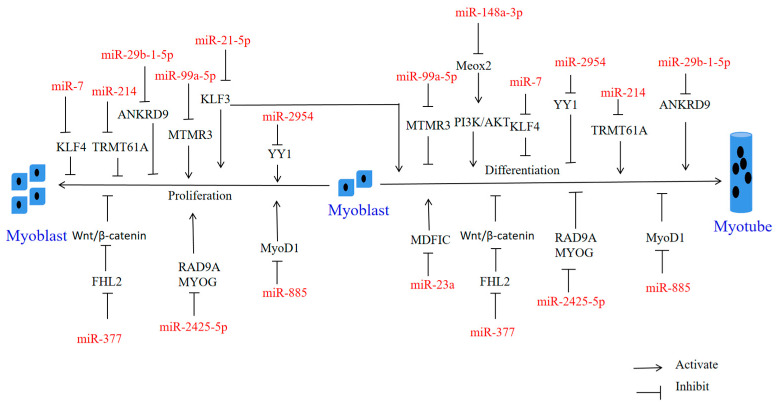
Regulation of cell proliferation and differentiation by miRNA in domestic chicken and bovine myoblasts. The upper half shows the myogenesis of miRNAs in domestic chickens and the lower half shows the myogenesis of miRNAs in cattle. miRNA is shown in red font, and signaling pathways and genes are shown in black font, and the names of the processes involved in skeletal muscle growth are shown in blue font.

**Table 1 ijms-24-14534-t001:** Types and functions of animal skeletal muscle-associated miRNAs.

Species	miRNA	Proliferation	Differentiation	Target	Source
Chicken	miR-99a-5p	Promote	Inhibition	MTMR3	[58]
Chicken	miR-21-5p	Promote	Promote	KLF3	[59]
Chicken	miR-148a-3p	No influence	Promote	Meox2	[60]
Chicken	miR-7	Inhibition	Inhibition	KLF4	[61]
Chicken	miR-2954	Promote	Inhibition	YY1	[62]
Chicken	miR-214	Inhibition	Promote	TRMT61A	[63]
Chicken	miR-29b-1-5p	Inhibition	Promote	ANKRD9	[64]
Bovine	miR-377	Inhibition	Inhibition	FHL2	[65]
Bovine	miR-23a	Unknown	Promote	MDFIC	[66]
Bovine	miR-2425-5p	Promote	Inhibition	RAD9A and MYOG	[67]
Bovine	miR-885	Promote	Inhibition	MyoD1	[68]
Sheep	miR-181a	Inhibition	Promote	YAP1	[69]
Sheep	miR-29a	Inhibition	Inhibition	Unknown	[70]
Goat	miR-99b-3p	Promote	Promote	Caspase-3 and NCOR1	[71]
Goat	miR-27a-3p	Unknown	Inhibition	ANGPT1	[72]
Goat	miR-145-5p	Unknown	Inhibition	USP13	[73]
Sheep	miR-22-3p	Inhibition	Promote	IGFBP3	[74]
Mouse	miR-424-5p	Promote	Inhibition	HSP90AA1	[75]
Mouse	miR-452	Promote	Inhibition	ANGPT1	[76]
Mouse	miR-424(322)-5p	Inhibition	Inhibition	Ezh1	[77]
Mouse	miR-495-3p	Inhibition	Promote	CDH2	[78]
Mouse	miR-22-3p	Inhibition	Promote	Unknown	[79]
Rabbit	miR-194-5p	Inhibition	Inhibition	Mef2c	[80]
Pig	miR-22	Inhibition	Promote	Unknown	[81]
Duck	miR-1 miR-133	No influence Promote	Promote No influence	HDAC4 SRF, TGFBR1	[82]

**Table 2 ijms-24-14534-t002:** Lnc RNAs in the regulation of skeletal muscle development.

Regulating Method	lncRNA	Ce RNA	Target Genes	Functions	Species	Source
Sponges miRNAs	lncRNA-MEG3	miR-133 a-3 p	PRRT 2	Regulation of the skeletal muscle regeneration	Mouse	[107]
	LncRNA H19	miR-140-5p	Drebrin 1	Inhibition of skeletal muscle satellite cell differentiation	Pig	[108]
	LncRNA H19	miR-20a-5p	TGFBR2	Promotes skeletal muscle fibrosis	Mouse	[109]
	lncMFAT1	miR-135a-5p	TGFBR135/SMAD5	Promotes skeletal muscle fibrosis	Mouse	[110]
	lncA2B1	miR-206	HNRNPA2B1	Promotes myogenic cell differentiation and myogenesis	Bovine	[111]
	lncR-133a	miR-133a-3p	FGFR1/ERK1/2	Inhibition of myogenic cell differentiation	Goat	[112]
Regulation of Gene expression in cis or trans	MUNC lncRNA		Myod1	Promotes myogenic cell differentiation	Mouse	[100]
	LncMyolinc		Filip1	Inhibits myoblast differentiation into myotubes	Mouse	[113]
	LncRNA-Six1		Six1	Promotes myoblast proliferation	Chicken	[114]

**Table 3 ijms-24-14534-t003:** Functions and regulatory mechanisms of circRNA in skeletal muscle development.

circRNA	Regulation Method	Function	Species	Source
circ-FoxO3	miR-2400 sponge	Inhibition of differentiation of myogenic cells	Mouse	[118]
circNfix	miR-204-5p sponge	Promotes myoblast differentiation	Mouse	[125]
circNDST1	miR-411a sponge	Promotes proliferation and inhibits cell differentiation of bovine myogenic cells	Bovine	[119]
circITSN2	miR-218-5p sponge	Promotion of skeletal muscle development in chicken embryos	Chicken	[120]
CircCCDC91	binding to the miR-15 family	Promotes myogenic proliferation and differentiation, and relieves skeletal muscle atrophy	Chicken	[121]
circUSP13	miR-29c sponge	Promotes differentiation and inhibit apoptosis of goat myogenic cells	Goat	[122]
CircUBE3A	miR-28-5p sponge	Promotes myogenic cell proliferation and differentiation	Goat	[123]
circIGF1R	miR-16 sponge	Promotes myogenic cell differentiation	Pig	[124]

## Data Availability

Not applicable.

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
