# Peer review of "The Function and Regulation Mechanism of Non-Coding RNAs in Muscle Development"

_ijms, 2023, doi:10.3390/ijms241914534_

Round 1
Reviewer 1 Report
The present article gives a nice overview of classes of non-coding RNAs and their involvement in muscle biology with the aim to highlight the importance of understanding non-coding RNAs to optimise meat production.
Major comment:
As I understand from reading this article non-coding RNAs and their function are often poorly conserved between species. Thus, it may not be useful to combine information on several species into 1 figure as has been done in figure 2. Unless of course these RNAs are in fact conserved?
Also, since this article focuses specifically on non-coding RNAs in livestock, it might be good to limit description of mouse and human non-coding RNAs or -perhaps, following a general section have a section specifically discussing livestock species and RNAs that may be interesting targets for the livestock industry.
Minor comments:
Line 50: "where progenitor cells undergo complex changes to form myoblasts and then differentiate to form myoblasts, which fuse to form myotubes."
Please review this sentence. “Myoblasts differentiate to form myoblasts” does not make sense to me. Perhaps this should say myogenic cells differentiate to form myoblasts?
Perhaps the illustration of a myofiber in figure 1 can be adjusted to indicate their much longer length compared to myotubes.
"Both miRNAs bound to the 3'-UTR and CDS regions are classically negatively regulated in the same way, while binding to the 5'-UTR mainly plays a transcriptional activation role[29]"
Are miRNAs that bind to the 3’UTR and CDS regions negatively regulated (meaning they themselves are suppressed) OR do miRNAs that bind to the 3’UTR and CDS regions suppress target gene expression as indicated in (29)? Please amend this sentence to correctly represent the literature.
Table 1: Perhaps it would be helpful to name the RNA type in the column “RNA type” (eg. ecircRNA, lncRNA, miRNA) and then add a column for the RNA name (eg circLRBA, lincRNA-UFC1, etc). At the moment the column heading “RNA type” is confusing as the authors are providing the name not the type.
Line 164-165: Remove comment: Table 1. This is a table. Tables should be placed in the
main text near to the first time they are cited.
Line 169: this sentence may be easier to understand if “such as” is replaced by “for example”.
Line 170-179: I am not sure what this extremely long sentence means. Please revise this sentence.
Section 4 reads predominantly as a list of non-coding RNAs and their function without telling much of a story. Much of 4.1 could be moved into the legend of Figure 2, which illustrates this section nicely. Some sentences are too long and confusing (eg. Line 170-179); occasionally gramma should be checked as well.
Line 249: FOR subsequent scholars
Line 274: In addition, circRNAs can interact with proteins to form RNA-protein complexes and thus function.
Is this sentence finished? Perhaps the authors are trying to say that circular RNAs can function by forming RNA-protein complexes?
Line 284: It is a bit strange to say skeletal muscle is closely related to meat production rates and meat quality. Perhaps it would make more sense to say that skeletal muscle growth and biology relates to meat production …?
Line 289: Different animals to which animals? Mice rather than cow? Can the authors please specify?
297: still MOSTLY? Unknown
Generally, this article is well written with appropriate use of the English language. Some sentences are long and some grammatical errors were present which can be easily addressed.
Author Response
Thank you very much for your comments and professional advice. These perspectives contribute to the academic rigor of our articles. Based on your suggestions and requests, we have made corrections to the revised manuscript. We want to display the following details:
The author's answer is in red. Due to the high number of revisions, revisions not answered in detail are reflected in the manuscript.
Reviewer 1.
As I learned from reading this article, non-coding RNAs and their functions are often poorly conserved between species. Therefore, it may not be useful to combine information from several species into a single figure as in Figure 1. Unless of course these RNAs are actually conserved?
We believe that non-coding RNAs lack strict sequence conservation between species but are conserved at multiple levels of sequence, RNA structure, genomic location and mechanism of action.
Furthermore, since this paper is specifically focused on non-coding RNAs in livestock, it would be better to limit the description to mouse and human non-coding RNAs, or - perhaps, have a section after the general section dedicated to livestock species and RNAs, which could be livestock species Interesting goals for the industry.
Description of mouse and human non-coding RNAs are reviewed.
Small comments:
Line 50: “Progenitor cells undergo complex changes to form myoblasts, which then differentiate to form myoblasts, which fuse to form myotubes.
Please review this sentence. "Myoblasts differentiate to form myoblasts" doesn't make sense to me. Maybe this should say myoblasts differentiate into myoblasts?
Modified: Myogenesis is divided into two phases. The first stage refers to the formation of myoblasts through complex changes in progenitor cells during embryonic development. These cells proliferate, migrate, and fuse to form multinucleated myotubes induced by specific myogenic transcription factors. The myotubes eventually fuse into muscle fibers.
Perhaps the illustration of muscle fibers in Figure 1 could be adjusted to show that they are much longer in length compared to myotubes.
already edited:
"MiRNAs that bind to the 3'-UTR and CDS regions are generally negative regulators in the same way, while those that bind to the 5'-UTR mainly play a role in transcriptional activation [29]" MiRNAs that bind to the 3'UTR and CDS regions are negative regulators (meaning they themselves are repressed), or do the miRNAs binding to the 3'UTR and CDS regions repress target gene expression, as shown in (29)? Please revise this sentence to correctly represent the literature.
Already modified: Mature miRNAs can mainly bind to complementary sites in the 3'-UTR region of target gene mRNA, triggering degradation or translational inhibition of target mRNA. Some miRNAs can also bind to 5'-UTR and coding regions, thereby silencing genes. Express.
Table 1: Maybe it would be helpful to name the RNA type (e.g. ecircRNA, lncRNA, miRNA) in the "RNA type" column and then add a column for the RNA name (e.g. circLRBA, lincRNA-UFC1, etc.). Currently, the column titled "RNA type" is confusing because the authors provide the name rather than the type.
already edited
Lines 164-165: Remove comment: Table 1. This is a table. Tables should be placed in the text, close to the time of the first citation.
already edited
Line 169: This sentence might be easier to understand if you replace "for example" with "for example."
already edited
Lines 170-179: I'm not sure what this extremely long sentence means. Please modify this sentence.
Already modified: For example, the stability and RNA secondary structure of LncRNA MEG3 can be altered by SNPs that affect the transcription of LncRNA MEG3. Specifically, two favorable haplotypes (MEG3-TTCC and MEG3-CCCA) developed in fat and lean pigs can inhibit satellite cell proliferation through inactivation of the PI3K/AKT and MAPK/ERK1/2 pathways , and promotes satellite cell differentiation through the JAK3/STAT3 pathway [86]; MUNC lncRNA acts as an enhancer RNA for the muscle growth-promoting factor Myod1 gene in a cis-regulatory manner, and can stimulate other muscle growth-promoting genes in trans by recruiting the clusterin complex expression, thereby regulating skeletal muscle growth [87]; actin nucleation factor (Spire1) is involved in the regulation of normal skeletal muscle differentiation, and lnc-SMaRT interacts with the G-quadruplex-containing region of Mlx-γ mRNA, thereby inhibiting Translation of Spire1 to further regulate skeletal muscle differentiation [88]. During cardiac and skeletal muscle development, Ppp1r1b lncRNA can promote myogenic differentiation by competing for chromatin binding of the myogenic regulatory polycomb repressive complex 2 (PRC2) gene with the myogenic master regulator. [89]
Section 4 mainly serves as a list of non-coding RNAs and their functions without telling much of the story. Much of Figure 4 can be moved into the legend of Figure 1, and Figure 2 illustrates this section well. Some sentences are too long and confusing (e.g. lines 170-179); grammar should be checked occasionally.
already edited
Line 249: For later scholars
already edited
Line 274: In addition, circRNA can interact with proteins to form RNA-protein complexes, thereby exerting their effects. Is this sentence finished? Maybe the authors are trying to say that circular RNAs can function by forming RNA-protein complexes?
Yes, the description means that circRNA can function by forming RNA-protein complexes.
Line 284: It is a bit strange to say that skeletal muscle is closely related to meat production and meat quality. Maybe it makes more sense to say that skeletal muscle growth and biology are related to meat production...?
Skeletal muscle is an important tissue of animals. The growth and biology of skeletal muscle are closely related to meat production. Therefore, fully understanding the role of non-coding RNA in the development of animal skeletal muscle can better control the growth rate and quality of livestock and poultry. and improve its economic benefits.
Line 289: Different animals versus which animals? Rat instead of cow? Can the author elaborate?
As described in Tables 2, 3, 4, and 4 of the article, Currently, researchers have identified a large number of ncRNAs that regulate skeletal muscle in animals such as pigs, cows, sheep, chickens and mouse.
297: still MOSTLY? Unknown
We believe that many non-coding RNAs are found in animals, but the life activities and specific mechanisms of action in which they are involved in the animal body are not clear.
Thank you very much for your attention and time. Look forward to hearing from you.
Your sincerely,
Yaling Yang
13·July,2023
Reviewer 2 Report
Dear Authors,
The regulatory function of non-coding RNA became of interest for many researchers in recent yeras. It would be interesting to see their function and possible mechanism of action in a comprehensive review. Authors undertake the effort to collect information about non-coding RNAs function in muscle development and proliferation. However the presentation of data is very poor. It is written with very basic language, including many repetitions. Abstract is non-informative where in very short form (6 lines) authors provide information that is several times repeated and could be shorten to 2 sentences. Similarly introduction including general view of muscle development is written with very basic language and in chaotic manner. There was probably a good idea to inroduce different classes of non-coding RNAs described in the review as authors tried to do in the paragraph 3. However, it is also very chaotic and includes not organized infromation. Table 1 included in this part is very general and there is information of diverse functions of diverse nc-RNA, which have nothing in common with muscles development or formation or function. Paragraph 4.1, which describes role of miRNAs in muscle function is simply a collection of information about several different miRNas. Each sentence is different miRNA. Authors did not try to organize it in a context dependent manner, mechanism or function (e.g based on figure 2 and Table 2 - which are OK, it could be divided into miRNAs with the role in proliferation, myoblast function etc). In some parts of the text there are authors comments left (e.g. line 163-165). The same situation is in part describing lcnRNA role. Line 168-170 is absoultly impossible to understand. Rest of this paragraph is just independent sentences put together. Table 3 referes only to the role of H19, which is not easy to understand why it was chosen and not all lncRNA. Table 4 and 5 looks to be fine. Circular RNA part is written in similar way as two others. It is chaotic, repetitive and combines loosely connected information about different circRNAs. Conclusions are also not very informative with many assumptions, without a good summary of the presented data. In general despite the interesting idea, and several good looking tables, and of course number of reviewed articles cited here, the quality of the work is very poor and need to be complitly reorganized and rewriten to create logic pattern.
English language is very basic, the sentences are formed very often in not logic order. They are long and contain many repetitions.
Author Response
Dear revlewer,
Thank you very much for your comments and profession aladvice. These opinions help to improve academic rigor of our article. Based on your suggestion and·request, we have made corrected modifications on the revised manuscript. we would like to show the details as follows:
Reviewer 2.
The regulatory function of non-coding RNA became of interest for many researchers in recent yeras. It would be interesting to see their function and possible mechanism of action in a comprehensive review. Authors undertake the effort to collect information about non-coding RNAs function in muscle development and proliferation. However the presentation of data is very poor. It is written with very basic language, including many repetitions. Abstract is non-informative where in very short form (6 lines) authors provide information that is several times repeated and could be shorten to 2 sentences. Similarly introduction including general view of muscle development is written with very basic language and in chaotic manner. There was probably a good idea to inroduce different classes of non-coding RNAs described in the review as authors tried to do in the paragraph 3. However, it is also very chaotic and includes not organized infromation. Table 1 included in this part is very general and there is information of diverse functions of diverse nc-RNA, which have nothing in common with muscles development or formation or function. Paragraph 4.1, which describes role of miRNAs in muscle function is simply a collection of information about several different miRNas. Each sentence is different miRNA. Authors did not try to organize it in a context dependent manner, mechanism or function (e.g based on figure 2 and Table 2 - which are OK, it could be divided into miRNAs with the role in proliferation, myoblast function etc). In some parts of the text there are authors comments left (e.g. line 163-165). The same situation is in part describing lcnRNA role. Line 168-170 is absoultly impossible to understand. Rest of this paragraph is just independent sentences put together. Table 3 referes only to the role of H19, which is not easy to understand why it was chosen and not all lncRNA. Table 4 and 5 looks to be fine. Circular RNA part is written in similar way as two others. It is chaotic, repetitive and combines loosely connected information about different circRNAs. Conclusions are also not very informative with many assumptions, without a good summary of the presented data. In general despite the interesting idea, and several good looking tables, and of course number of reviewed articles cited here, the quality of the work is very poor and need to be complitly reorganized and rewriten to create logic pattern.
Author's Answer.:
First, changes were made to the abstract:Non-coding RNAs are involved in regulating the proliferation, differentiation and apoptosis of animal skeletal muscle myogenic cells. This paper focuses on the mechanisms by which miRNAs, lncRNAs and circRNAs regulate the growth and development of animal skeletal muscle, aiming to provide reference for the in-depth study of the role of ncRNAs in the development of animal skeletal muscle, and to provide new ideas for the improvement of animal production performance.Then, we mainly want Table I to reflect that the three RNAs are widely involved in a variety of important life activities in animals, including organ growth and development, cell proliferation and differentiation, glycolipid metabolism and disease aspects, etc. For Figure 2, we considered. The description of lncRNAs was removed from the figure, leaving only the miRNA ones. Meanwhile Table III was expanded with descriptions, which might make the article a bit clearer. Finally, the incomprehensible statements and incoherent places in the text were modified. The changes are rather trivial and disorganized and thus not listed in the response, but are reflected in the uploaded manuscript.
Thank you very much for your attention andtime. Look forward to hearing from you.
Your sincerely,
Yaling Yang
13·July,2023
Round 2
Reviewer 2 Report
I would say that in general the review combines many information, if someone would try hardly, it could be possible to obtain some needed data form this review. However it is not a case in writing a review. The data should be organized, collected and presented in logic, easy to read form. And this review is not such a presentation. I do not see too much improvement of the quality of presented information. Quality is still poor, very basic language. Information is chaotic not organised. It is neede to re-organize the whole structure of article and not only make minimal manipulation in the text. I am opting to reconsider it after major corrections.
In the abstract we have in first sentence "animal skeletal muscle myogenic cells" in second sentence "growth and development of animal skeletal muscle" and in the same very long sentence again "role of ncRNAs in the development of animal skeleteal muscle".
First lines of Introduction (19-21) - several times repeated important part, important role, animal body. When the readers go further into the lecture of this manuscript there is the same notion that very basic is used. Especially statments such as "study the role.... by overexpressing or suppressing the expression on non-coding RNAs" or "some researchers constucted mouse models". This would be much more professional to say - Many in vitro studies including overexpression and knocout has been applied to study the role of non-coding RNAs. Additionaly mouse models helped to better understand their role in muslce development.
Lines 33-35 - repetitions of the same words, or formualtions "key regulators of animal muscle development, the understanding of ncRNAs in the process of skeletal muscle growth and development…"
Line 41 – it doesn’t make sense to say „Animal muscle development is important for both production and economy” and combine this with the information in the same sentence to say „ there are three main types of animal muscle are cardiac, smooth and skeletal muscle”. Thoe information do not fit together in such way. Line 42-43 – what does it even mean that the function of skeletal muscle depends on „the growth and differentiation of myotubes, the transformation of muscle fiber types and mitochondrial function”? - IT would sound totally different to say „Organized and ordered process of developemnt and differentiation of skeletal muscles determines their function. Due to very high energetic demands, muscles are characterized by high metabolic rates where mitochondria playes crucial role.”
Lines 57-72 contains number of transcription factors, which could be described in more interesting way. Authors totally do not take advantage of the Figure 1 that they provided. This would be much better idea to re-organize description in the order of factors and processes. Start from stem cells that proliferate due to the action of PAX3 and 7. Furhter activation occures thanks to the role of some myoG factors, Myoblast and myotubes fully developed components of muscles dependes on the action of othe factors, etc.
Furhter to the description of miR function. Line 126-127 „During animal skeletal muscle development, a large number of miRNAs have been shown to influence the proliferation and differentiation of animal skeletal muscle cells and are important regulators of development.”. It is merely a repetition of the same phrase in one sentence. The rest major part of the main core of this article which should be description of the ncRNA role I just tried to read and could not make any comment. Nothing improved from the previous version, maybe some additional explanataions added, but there is no ordere, no connection between described processes. On miRNA inhibits this, other activates something. Many of them act together, or in order. Nothing is reflected in the authors manuscript. It is collection of information about several ncRNAs. Morover sentences such as „The role of miRNAs in the growth and development of skeletal muscle is becoming more and more prominent, but the specific mechanisms involved in the regulation of skeletal muscle growth and development still need to be further understood, therefore, more research efforts are needed in this area.” is totally redundant and add no quality to the text. The same is about role of lncRNA and circRNAs
Basic language and sometimes, long sentences that are not easy to follow. Many sentences are just mentioned written without much connection to the rest of the text
Author Response
Point 1:
I would say that in general the review combines many information, if someone would try hardly, it could be possible to obtain some needed data form this review. However it is not a case in writing a review. The data should be organized, collected and presented in logic, easy to read form. And this review is not such a presentation. I do not see too much improvement of the quality of presented information. Quality is still poor, very basic language. Information is chaotic not organised. It is neede to re-organize the whole structure of article and not only make minimal manipulation in the text. I am opting to reconsider it after major corrections.
Response 1: First of all, thank you very much for your valuable suggestions, has been based on the recommendations of the line of major changes
Point 2:
In the abstract we have in first sentence "animal skeletal muscle myogenic cells" in second sentence "growth and development of animal skeletal muscle" and in the same very long sentence again "role of ncRNAs in the development of animal skeleteal muscle".
Response 2: Abstract has been modified to reduce duplication in skeletal muscle growth and development
Point 3:
First lines of Introduction (19-21) - several times repeated important part, important role, animal body. When the readers go further into the lecture of this manuscript there is the same notion that very basic is used. Especially statments such as "study the role.... by overexpressing or suppressing the expression on non-coding RNAs" or "some researchers constucted mouse models". This would be much more professional to say - Many in vitro studies including overexpression and knocout has been applied to study the role of non-coding RNAs. Additionaly mouse models helped to better understand their role in muslce development.
Response 3: Reductions have been made on“several times repeated important part, important role, animal body” Repeat. And as suggested the sentence has been changed to Many in vitro studies, including overexpression and knockdown, have been applied to study the role of noncoding RNAs; In addition, mouse models help to better understand their role in muscle development.
Point 4:
Lines 33-35 - repetitions of the same words, or formualtions "key regulators of animal muscle development, the understanding of ncRNAs in the process of skeletal muscle growth and development…"
Response 4:Already in place as recommended.
Point 5:
Line 41 – it doesn’t make sense to say „Animal muscle development is important for both production and economy” and combine this with the information in the same sentence to say „ there are three main types of animal muscle are cardiac, smooth and skeletal muscle”. Thoe information do not fit together in such way. Line 42-43 – what does it even mean that the function of skeletal muscle depends on „the growth and differentiation of myotubes, the transformation of muscle fiber types and mitochondrial function”? - IT would sound totally different to say „Organized and ordered process of developemnt and differentiation of skeletal muscles determines their function. Due to very high energetic demands, muscles are characterized by high metabolic rates where mitochondria playes crucial role.”
Response 5: Appropriate cuts were made to words that did not make sense.
“the growth and differentiation of myotubes, the transformation of muscle fiber types and mitochondrial function”This implies that the process of myogenesis requires the coordination of multiple factors to control the activation of quiescent satellite cells, proliferation of adult myoblasts, exit from the cell cycle, and subsequent terminal differentiation leading to multinucleated myofibers
Point 6:
Lines 57-72 contains number of transcription factors, which could be described in more interesting way. Authors totally do not take advantage of the Figure 1 that they provided. This would be much better idea to re-organize description in the order of factors and processes. Start from stem cells that proliferate due to the action of PAX3 and 7. Furhter activation occures thanks to the role of some myoG factors, Myoblast and myotubes fully developed components of muscles dependes on the action of othe factors, etc.
Response 6: We have some description of the role of the mechanism later, which may explain it somewhat. As in 4.1.
Point 7:
Furhter to the description of miR function. Line 126-127 „During animal skeletal muscle development, a large number of miRNAs have been shown to influence the proliferation and differentiation of animal skeletal muscle cells and are important regulators of development.”. It is merely a repetition of the same phrase in one sentence. The rest major part of the main core of this article which should be description of the ncRNA role I just tried to read and could not make any comment. Nothing improved from the previous version, maybe some additional explanataions added, but there is no ordere, no connection between described processes. On miRNA inhibits this, other activates something. Many of them act together, or in order. Nothing is reflected in the authors manuscript. It is collection of information about several ncRNAs. Morover sentences such as „The role of miRNAs in the growth and development of skeletal muscle is becoming more and more prominent, but the specific mechanisms involved in the regulation of skeletal muscle growth and development still need to be further understood, therefore, more research efforts are needed in this area.” is totally redundant and add no quality to the text. The same is about role of lncRNA and circRNAs
Response 7: Role descriptions of mirRNAs, lncRNAs and circRNAs have been categorized and re-edited. The modifications are not explained in the reply as I think the changes are large. I apologize for any inconvenience this may cause to reviewers.
Round 3
Reviewer 2 Report
Dear authors,
I believe that with all the effort to improve the manuscript there are some important aspects missing. First of all, I believe that professional english correction must be applied. IJMS journal with its high impact factor should maintain quality articles. Therefore some additional work on this manuscript must be done. However, I believe that those changes to current version might be promissing for the publication. There are some better written parts that need minor inspections. However some paragraphs still need some more work and considerations. Please see some comments that may improve your manuscript.
Starting with introduction, there are still repetitions such as "With the identification of a large number of non-coding RNAs as key regulators of muscle development in animals, there has been a gradual increase in the understanding of non-coding RNAs in skeletal muscle growth and development" (line 31-34). - this could be simplified and shorthened such as - Identification of a large number of non-coding RNAs as key regulators of muscle growth and development in animals has been gradually increasing our understanding of their role. Similar situation line 35-37 - " This review focuses on the regulation of animal skeletal muscle development by miRNAs, lncRNAs and circRNAs, aiming to provide a deeper understanding of the mechanisms by which non-coding RNAs regulate the growth and development of animal skeletal muscle".
Introduction including muscle development is still enigmatic and could be improved for better understanding. Part where authors explain basics of non-coding RNAs is significantly improved with good idea to include Fig 2.
Line 147 - " A single miRNA can be regulated by multiple mRNAs, and a single mRNA can likewise target multiple miRNAs". miRNA should be re-written to mRNA and mRNA switched to miRNA
Line 148-154 should be shortened and bound togehter to form 1 sentence, maximum 2. Those sentences are merely repetition without introduction of significant information.
Lines 164-169 - There is too much general information. miRNA regulates transcription factors, TF vice versa regulate miRNA, and miRNA regulates genes. This should be shortened and presented much clearly, with 1 max 2 general descriptisons as further below there are specific roles of miRNAs mentioned.
There is lack of clarity " In contrast, double knockout of miR-499 and miR-208b in mice resulted in significant deletion of type I fibers in flounder muscle" - is this mice or flounder? And also in few places in the text authors indicate role of ncRNA in poultry and live stock production however you also mention fish, which creates some discrepancy (lines 252-253).
In general there is significant improvment in presentation of data, but still some pargraphs such as paragraph 4.2.1 or 4.2.2 are just examples introduced one by one.
Advicing professional english correction.
Author Response
Dear Editor and dear reviewer:
Thank you for your letter and the reviewers for their comments on our manuscript. These opinions are valuable and helpful. We have carefully read all comments and made corrections. According to the instructions provided in your letter, we have uploaded the revised draft file. Where revisions have been made in the text, they have been highlighted in red, and responses to reviewers have been highlighted in red and shown below.
Point 1: Starting with introduction, there are still repetitions such as "With the identification of a large number of non-coding RNAs as key regulators of muscle development in animals, there has been a gradual increase in the understanding of non-coding RNAs in skeletal muscle growth and development" (line 31-34). - this could be simplified and shorthened such as - Identification of a large number of non-coding RNAs as key regulators of muscle growth and development in animals has been gradually increasing our understanding of their role. Similar situation line 35-37 - " This review focuses on the regulation of animal skeletal muscle development by miRNAs, lncRNAs and circRNAs, aiming to provide a deeper understanding of the mechanisms by which non-coding RNAs regulate the growth and development of animal skeletal muscle".
Response 1: As suggested, it has been amended to read: Identification of a large number of non-coding RNAs as key regulators of muscle growth and development in animals has been gradually increasing our understanding of their role. This review focuses on the regulation of animal skeletal muscle development by miRNAs, lncRNAs and circRNAs, aiming to provide a deeper understanding of the mechanisms by which non-coding RNAs regulate the growth and development of animal skeletal muscle.
Point 2: Line 147 - " A single miRNA can be regulated by multiple mRNAs, and a single mRNA can likewise target multiple miRNAs". miRNA should be re-written to mRNA and mRNA switched to miRNA
Response 2: As suggested, it has been modified to:“A single mRNA can be regulated by multiple miRNAs, and a single miRNA can likewise target multiple mRNAs”
Point 3: Line 148-154 should be shortened and bound togehter to form 1 sentence, maximum 2. Those sentences are merely repetition without introduction of significant information.
Response 3: As suggested, it has been modified to:“In recent years, miRNAs have played important roles in animal skeletal muscle development, including regulation of myoblast proliferation, differentiation, fusion, apoptosis and fiber type specification . Among them, most miRNAs mainly affect the proliferation and differentiation of myoblasts and satellite cells (see Table 1 and Figure 3).”
Point 4: Lines 164-169 - There is too much general information. miRNA regulates transcription factors, TF vice versa regulate miRNA, and miRNA regulates genes. This should be shortened and presented much clearly, with 1 max 2 general descriptisons as further below there are specific roles of miRNAs mentioned.
Response 4: As suggested, it has been simplified to:“These transcription factors can act synergistically or antagonistically with each other. miRNAs can target multiple myogenesis transcription factors and thus affect myogenesis; at the same time, their expression is directly controlled by these transcription factors [83, 84]. This relationship between miRNAs and myogenesis regulatory factors supports that miRNAs are integral to myogenesis regulatory network components.”
Point 5: There is lack of clarity " In contrast, double knockout of miR-499 and miR-208b in mice resulted in significant deletion of type I fibers in flounder muscle" - is this mice or flounder? And also in few places in the text authors indicate role of ncRNA in poultry and live stock production however you also mention fish, which creates some discrepancy (lines 252-253).
Response 5: Thank you very much for your advice. Due to my unclear writing and expression, the meaning is wrong. This refers to the leg muscles of mice (soleus).
Has been modified correctly: "In mouse skeletal muscle, overexpression of miR-499 can completely transform fast muscle fibers of soleus muscle into slow muscle fibers . In contrast, double knockout of miR-499 and miR-208b resulted in a significant loss of type I fibers in the soleus .”
Point 6: In general there is significant improvment in presentation of data, but still some pargraphs such as paragraph 4.2.1 or 4.2.2 are just examples introduced one by one.
Response 6: The following changes have been made to 4.2.1 and 4.2.2:
4.2.1
Emerging studies have shown that lncRNAs can regulate muscle proliferation and differentiation by competing for endogenous RNAs. lncRNA-MEG3 can act as a molecular sponge for miR-133 a-3 p to regulate the expression level of PRRT 2, thereby modulating skeletal muscle regeneration [107]. In pigs, lncRNA H19 can act as a molecular sponge for miR-140-5p to inhibit porcine skeletal muscle satellite cell differentiation and can also bind directly to Drebrin 1 to regulate satellite cell differentiation [108]. In mice, lncRNA H19 can regulate TGFBR2 expression through endogenous competing RNA functions enabling miR-20a-5p to activate the TGFβ/Smad pathway and promote skeletal muscle fibrosis in mice [109]. Furthermore, lncMFAT1 can act as a miR-135a-5p sponge to activate the TGFBR135 / SMAD5 pathway and promote skeletal muscle fibrosis in mice [110]. In cattle, lncA2B1 can regulate HNRNPA2B1 expression through endogenous competing RNA functions enabling miR-206 to promote bovine myoblast differentiation and myogenesis [111]. miR-133a-targeted GosB is sponged by muscle differentiation-associated lncRNA (MDNCR) to promote differentiation of bovine primary myoblasts [115]. In goats and sheep, lncR-133a acts as a miR-133a-3p sponge, activates the FGFR1 / ERK1 / 2 pathway in goats, and inhibits myogenic cell differentiation [112]. lncRNA (CTTN-IT1) can target miR-29a to act as a competing endogenous RNA for YAP1 and promote the proliferation and differentiation of skeletal muscle satellite cells in hu sheep [116].
4.2.2
In mice, lncMyolinc controls the expression of the protein-coding gene Filip1 in a cis manner, and knockdown of lncMyolinc and Filip1 inhibits the differentiation of myoblasts into myotubes, and Myolinc binds to the TAR DNA-binding protein (TDP-43) during myogenesis, resulting in the expression of muscle-specific genes [113]. In the chicken, lncRNA- Six1 overexpression enhances the expressi on of muscle growth-related genes (MYOG, MYHC, MYOD, IGF1R, and INSR) and encodes a micropeptide that affects the expression of the Six1 protein in a cis-regulatory manner and promotes the proliferation of myogenic cells [114]. In bovine, lnc403 negatively regulates the expression of the neighboring gene Myf6 and can positively regulate the expression of the interacting protein KRAS, inhibiting bovine skeletal muscle satellite cell differentiation[117].